# Deletion of ribosomal protein genes is a common vulnerability in human cancer, especially in concert with *TP53* mutations

Ram Ajore[1,†], David Raiser[2,†], Marie McConkey[2], Magnus Jöud[1], Bernd Boidol[2], Brenton Mar[2], Gordon Saksena[3], David M Weinstock[4], Scott Armstrong[5], Steven R Ellis[6], Benjamin L Ebert[2,3,*,‡] & Björn Nilsson[1,3,‡,**] (iD)

## Abstract

Heterozygous inactivating mutations in ribosomal protein genes (RPGs) are associated with hematopoietic and developmental abnormalities, activation of p53, and altered risk of cancer in humans and model organisms. Here we performed a large-scale analysis of cancer genome data to examine the frequency and selective pressure of RPG lesions across human cancers. We found that hemizygous RPG deletions are common, occurring in about 43% of 10,744 cancer specimens and cell lines. Consistent with p53-dependent negative selection, such lesions are underrepresented in *TP53*-intact tumors ($P \ll 10^{-10}$), and shRNA-mediated knockdown of RPGs activated p53 in *TP53*-wild-type cells. In contrast, we did not see negative selection of RPG deletions in *TP53*-mutant tumors. RPGs are conserved with respect to homozygous deletions, and shRNA screening data from 174 cell lines demonstrate that further suppression of hemizygously deleted RPGs inhibits cell growth. Our results establish RPG haploinsufficiency as a strikingly common vulnerability of human cancers that associates with *TP53* mutations and could be targetable therapeutically.

**Keywords** cancer; ribosomal gene haploinsufficiency; ribosome function
**Subject Categories** Cancer; Genetics, Gene Therapy & Genetic Disease

## Introduction

The human ribosome consists of an rRNA scaffold and about 80 proteins, divided into two subunits. Ribosomal protein genes (RPGs) are denoted *RPS1*, *RPS2*, etc. for the small (40S) subunit; *RPL1*, *RPL2*, etc. for the large (60S) subunit.

Several recent lines of evidence hold that mutation of RPGs leads to specific clinical and cellular phenotypes. Germline heterozygous inactivating mutations or deletions in *RPS19* and at least eight other RPGs cause Diamond-Blackfan anemia (DBA), a disorder characterized by macrocytic anemia and cancer predisposition, and the founding member of a class of disorders known as ribosomopathies (Draptchinskaia *et al*, 1999; Gazda *et al*, 2006, 2008, 2012; Farrar *et al*, 2008; Narla & Ebert, 2010; Vlachos *et al*, 2012; Raiser *et al*, 2014). Acquired somatic mutations in specific RPGs associate with certain malignancies, including the 5q- subtype of myelodysplastic syndrome (hemizygous deletions of *RPS14*; Ebert *et al*, 2008), T-cell acute lymphoblastic leukemia (mutations in *RPL5*, *RPL10*, and *RPL22*; Rao *et al*, 2012; De Keersmaecker *et al*, 2013), and microsatellite-unstable endometrial and gastric cancer (mutations in *RPL22*; Nagarajan *et al*, 2012; Novetsky *et al*, 2013). Eleven RPGs are tumor suppressor genes in zebrafish, where hemizygous inactivation of these genes causes malignant peripheral nerve sheath tumors with high penetrance (Amsterdam *et al*, 2004; MacInnes *et al*, 2008; Lai *et al*, 2009). In mouse models, RPG haploinsufficiency alters stem cell quiescence (Signer *et al*, 2014), homeobox gene translation, and tissue patterning (Kondrashov *et al*, 2011; Xue *et al*, 2015).

Despite these observations, the occurrence of RPG lesions in human cancers has not been investigated systematically. We therefore carried out a large-scale analysis of cancer genome data to

1 Hematology and Transfusion Medicine, Department of Laboratory Medicine, Lund University, Lund, Sweden
2 Department of Medicine, Brigham and Women's Hospital, Harvard Medical School, Boston, MA, USA
3 Broad Institute, 7 Cambridge Center, Cambridge, MA, USA
4 Dana-Farber Cancer Institute, Boston, MA, USA
5 Memorial Sloan Kettering Cancer Center, New York, NY, USA
6 Department of Biochemistry and Molecular Biology, University of Louisville, Louisville, KY, USA
*Corresponding author. Tel: +1 617 355 9060; E-mail: benjamin_ebert@dfci.harvard.edu
**Corresponding author. Tel: +46 46 2220738; E-mail: bjorn.nilsson@med.lu.se
†These author contributed equally as first authors
‡These author contributed equally as last authors

determine the frequency and selective pressure of RPG lesions across human cancers. We first looked for chromosomal deletions and amplifications and point mutations in RPGs using pre-existing DNA copy number microarray and whole-exome sequencing data from a total of 10,744 cancer specimens and cell lines. Because recent studies have shown that RPG haploinsufficiency activates p53 in ribosomopathies (Narla & Ebert, 2010; Raiser *et al*, 2014) and the pathobiology of ribosomopathies can be alleviated by p53 inhibition (Volarevic *et al*, 2000; Sulic *et al*, 2005; Fumagalli *et al*, 2009; Barlow *et al*, 2010; Dutt *et al*, 2011; Jaako *et al*, 2011), we hypothesized that inactivation of RPGs could lead to negative selection unless the cells have mutated *TP53*, and, accordingly, looked for associations between inactivating RPG lesions and *TP53* mutation. While we observed few point mutations and homozygous deletions in RPGs, we detected hemizygous RPG deletions in a large proportion (43%) of samples. Consistent with negative selection, further analyses revealed an underrepresentation of RPG deletions in *TP53*-intact tumors, whereas we did not see any signs of negative selection in *TP53*-mutant tumors. Furthermore, functional experiments showed that deficiency of frequently deleted RPGs increases p53 activity in *TP53*-intact cell lines and perturbs rRNA maturation both in cell lines cultured *ex vivo* and in primary acute leukemia cells with specific RPG deletions and expanded *in vivo* in xenograft models. Finally, consistent with the low frequency of homozygous deletion, analysis of genomewide shRNA screening data showed that further suppression of hemizygously deleted RPGs inhibits the growth of RPG-haploinsufficient cancer cells. We conclude that RPG haploinsufficiency as a common feature of human cancers that associates with *TP53* mutations and could be targetable therapeutically.

# Results

## Haploinsufficiency for RPGs on the basis of hemizygous regional deletions is a common feature of cancer genomes

We first analyzed DNA copy number profiles of 7,225 cancer specimens belonging to 24 tumor types using data from The Cancer Genome Atlas, TCGA (Cancer Genome Atlas Research Network *et al*, 2013). Using $\log_2$ copy number ratio thresholds between $-0.3$ and $-0.7$ (median $-0.5$), we detected deletions affecting RPGs in 67–22% of specimens (median 43%), and in all tumor types analyzed (Fig 1A and Table 1). In 58–12% (median 32%) of tumors, we detected multiple RPG deletions (Fig 1C). Consistent with negative selection, we observed lower deletion frequencies for RPG than for other genes (Fig 1C and D). Validation in two independent sets of copy number profiles of 2,476 specimens belonging to 13 tumor types from Tumorscape (Beroukhim *et al*, 2010) and 1,043 cancer cell lines from the Cancer Cell Line Encyclopedia (CCLE; Barretina *et al*, 2012) identified the same RPGs as frequently deleted (Figs 1B and EV1, and Table 1). In all three data sets, almost all (97–100%) of RPG deletions had copy numbers consistent with hemizygous loss (Fig 2A), and few RPGs showed recurrent homozygous deletions (Table EV1). Throughout, RPGs were hit by regional deletions covering multiple genes, and all frequently deleted RPGs were co-deleted with well-known tumor suppressor genes, including *RPL26* on chromosome 17p, which is always co-deleted with *TP53*, and *RPS6* on chromosome 9p, which is always co-deleted with

*CDKN2A*. In addition to copy number changes, we examined the spectrum of point mutations in RPGs using whole-exome sequencing data from 4,655 TCGA samples. In contrast to the high frequency of hemizygous deletions, we found a low frequency of point mutations in RPGs, although we noted recurrent mutations in *RPL5*, *RPL10*, and *RPL22* in several tumor types as reported previously (Nagarajan *et al*, 2012; Rao *et al*, 2012; De Keersmaecker *et al*, 2013; Novetsky *et al*, 2013; Table EV2). In summary, these data show that RPG haploinsufficiency on the basis of regional deletions occurs in a large proportion of human cancers.

## Deletion of RPGs in cancer cells is restricted by p53-dependent negative selection

Recently, studies aimed at understanding the molecular mechanisms of ribosomopathies have identified p53 as a central mediator of the clinical features of these diseases (Narla & Ebert, 2010; Raiser *et al*, 2014). In both DBA and the 5q- syndrome, the hematopoietic phenotype is at least partly linked to p53 activation, and animal models have confirmed p53 as a sensor of ribosome dysfunction (Volarevic *et al*, 2000; Sulic *et al*, 2005; Fumagalli *et al*, 2009; Barlow *et al*, 2010; Dutt *et al*, 2011; Jaako *et al*, 2011). In normal cells, the expression of RPGs is tightly coordinated. In DBA and the 5q-syndrome, however, the RPG haploinsufficiency is thought to perturb the stoichiometry of ribosomal proteins, leading to inefficient ribosome assembly and increased concentrations of free ribosomal proteins, some of which (RPL5 and RPL11) regulate key components of the 5S ribonucleoprotein particle (Macias *et al*, 2010; Donati *et al*, 2013; Sloan *et al*, 2013; Goudarzi & Lindstrom, 2016). When ribosome biogenesis is blocked, the 5S SNP pre-ribosomal complex is re-directed from assembly into 60S ribosomes to MDM2 E3 ubiquitin ligase inhibition, thereby inhibiting the ability of MDM2 to target p53 for proteasomal degradation (Fumagalli *et al*, 2009; Zhang & Lu, 2009; Deisenroth & Zhang, 2010; Miliani de Marval & Zhang, 2011; Zhou *et al*, 2013; Goudarzi & Lindstrom, 2016). Moreover, *in vitro* and *in vivo* studies support that the phenotypic effects of RPG haploinsufficiency can be alleviated by genetic or pharmacological inhibition of p53 (Volarevic *et al*, 2000; Sulic *et al*, 2005; Fumagalli *et al*, 2009; Barlow *et al*, 2010; Dutt *et al*, 2011; Jaako *et al*, 2011).

Because of these reports, we hypothesized that acquisition of RPG deletions in cancer cells could lead to p53 activation and thereby negative selection, unless the p53 pathway has been inactivated. To test this hypothesis, we compared the copy number distributions for RPGs versus other genes across 4,675 TCGA samples having both copy number microarray and whole-exome sequence data, enabling the detection of RPG deletions as well as deletions and point mutations in *TP53*. We found an underrepresentation of hemizygous RPG deletions in *TP53*-wild-type tumors (Wilcoxon rank-sum $P \ll 10^{-10}$), whereas RPGs and other genes showed identical copy number distributions in *TP53*-mutant tumors (Fig 2A). Similarly, among *TP53*-wild-type tumors, but not among *TP53*-mutant tumors, we found fewer cases with hemizygous deletions affecting RPGs than with hemizygous deletions affecting random gene sets of the same size (permutation testing $P = 0.04$ with 1,000 random sets of genes located on other autosomes than chromosome 17), whereas no difference was detected in *TP53*-mutant tumors. Additionally, *TP53*-mutant tumors showed higher numbers of

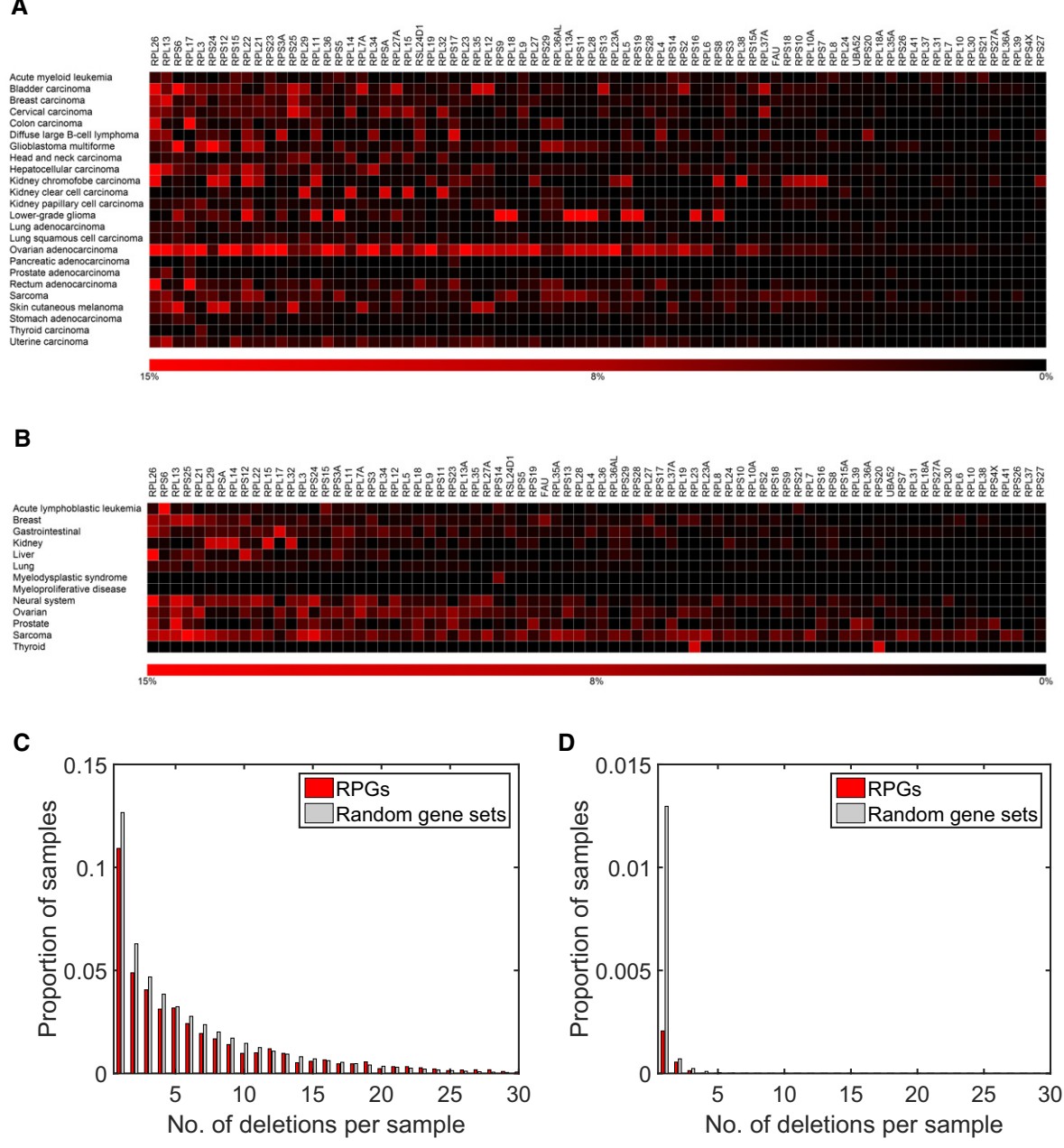

**Figure 1. Hemizygous deletion of RPGs is a common feature of cancer genomes.**

Given the evidence that RPG haploinsufficiency can alter cellular phenotypes and modulate oncogenesis, we performed a systematic examination of somatic alterations of RPGs in cancer broadly.

A    Deletion frequencies for different RPGs estimated from DNA copy number profiles of 7,275 primary cancer specimens belonging to 24 different tumor types (red).

B    Analysis of DNA copy number profiles of 2,476 primary cancer specimens belonging to 13 tumor types (including four not represented in the TCGA) identified the same RPGs as frequently deleted. Similar results were also calculated with DNA copy number profiles of 1,043 cancer cell lines (Fig EV1). The results shown were obtained using $\log_2$ ratio $-0.5$ as threshold. Similar results were obtained with other reasonable thresholds.

C    Histogram of number of deletions per tumor for RPGs and for 1,000 equally sized random gene sets in the TCGA samples. Tumors with multiple RPG deletions were common. Consistent with negative selection, we also detected fewer deletions in RPGs than in 1,000 random gene sets of the same size. This plot was obtained with a $\log_2$ threshold of $-0.5$, detecting both hemizygous and homozygous deletions.

D    Corresponding plot obtained with the more conservative $\log_2$ threshold $-2.0$, primarily detecting homozygous deletions.

RPG deletions in total (Fig 2B), RPG deletions that could be detected in *TP53*-wild-type tumors were enriched in tumors harboring alternative p53 pathway-inactivating lesions, including *MDM2* amplifications and *CDKN2A* deletions (Fig EV2). Taken together, these results indicate that the p53 pathway restricts the acquisition of RPG deletions in cancer cells.

**Table 1.  Ten most frequently deleted RPGs.**

| Gene | Chr | Start (bp) | End (bp) | TCGA | | CCLE | | Tumorscape | |
|------|-----|-----------|----------|------|------|------|------|------|------|
| | | | | *n* | % | *n* | % | *n* | % |
| *RPL26* | 17 | 8,280,833 | 8,286,565 | 1,752 | 24.2 | 340 | 32.6 | 310 | 12.5 |
| *RPL13* | 16 | 89,627,064 | 89,633,237 | 1,430 | 19.8 | 204 | 19.6 | 222 | 9.0 |
| *RPS6* | 9 | 19,376,253 | 19,380,235 | 1,411 | 19.5 | 373 | 35.8 | 269 | 10.9 |
| *RPL17* | 18 | 47,014,850 | 47,018,935 | 1,335 | 18.5 | 393 | 37.7 | 173 | 7.0 |
| *RPL29* | 3 | 52,027,643 | 52,029,958 | 1,283 | 17.8 | 340 | 32.6 | 229 | 9.2 |
| *RPL3* | 22 | 39,708,886 | 39,715,670 | 1,205 | 16.7 | 263 | 25.2 | 152 | 6.1 |
| *RPL14* | 3 | 40,498,782 | 40,503,863 | 1,133 | 15.7 | 292 | 28.0 | 209 | 8.4 |
| *RPSA* | 3 | 39,448,203 | 39,454,032 | 1,129 | 15.6 | 284 | 27.2 | 202 | 8.2 |
| *RPL21* | 13 | 27,825,691 | 27,830,702 | 1,121 | 15.5 | 328 | 31.4 | 224 | 9.0 |
| *RPS15* | 19 | 1,438,362 | 1,440,492 | 1,108 | 15.3 | 264 | 25.3 | 170 | 6.9 |

### Deletion of RPGs leads to loss of coordinated RPG expression and p53 activation

Previous studies aimed at understanding the molecular mechanisms of DBA and the 5q- syndrome have shown that decreased expression of specific RPGs (*RPS6*, *RPS14*, and *RPS19*) leads to p53 activation (Narla & Ebert, 2010; Raiser *et al*, 2014). However, the RPGs we found to be deleted in human cancers have not been studied functionally in this regard. To examine whether p53 senses decreased expression of the frequently deleted RPGs, we used shRNA-mediated knockdown in the *TP53*-wild-type lung adenocarcinoma cell line A549 (Fumagalli *et al*, 2009; Dutt *et al*, 2011). For all eight RPGs tested, multiple shRNAs caused elevated p53 protein levels and increased *P21* transcript levels (Figs 2C and D, and EV3, and Table EV3). Similarly, knockdown of three of the most frequently deleted RPGs in *TP53*-wild-type MOLM13 leukemia cells caused elevated expression of p53 and four p53 target genes *P21*, *BAX*, *PUMA*, and *NOXA* (Figs EV4 and EV5). Taken together, these data indicate that p53 activation results not only from decreased expression of the RPGs that are mutated in ribosomopathies, but also from decreased expression of the RPGs that are frequently deleted in cancer cells.

Since RPG haploinsufficiency is thought to activate p53 by altering ribosomal protein stoichiometry (Fumagalli *et al*, 2009), we tested for associations between RPG copy number anomalies (RPG-CNAs) and altered RPG expression patterns using 4,919 TCGA samples having both DNA copy number microarray data and global mRNA-sequencing data. We found a correlation between copy number and expression for most individual RPGs (Fig EV6A). Quantifying RPG co-expression by calculating correlation coefficients for all pairs of RPGs, we detected less correlated RPG expression in tumors with a high RPG-CNA burden (Fig EV6B and C), whereas tumors with few RPG-CNAs showed RPG co-expression patterns comparable to those observed in noncancerous tissues (Fig EV6D). These data indicate that RPG-CNAs lead to uncoordinated RPG expression, which, together with the shRNA knockdown data, provides a possible explanation for the negative selection of RPG deletion observed in *TP53*-wild-type tumors (Fig 2).

### Deletion of RPGs in cancer cells with intact p53 function

We asked which RPG deletions are permissible in cancer cells with intact p53 function. To this end, we specifically analyzed the 30 cell lines from CCLE that are both *TP53*-wild-type and sensitive to the p53-activating compound Nutlin-3 ($IC_{50} < 8$ μmol; Barretina *et al*, 2012; Sonkin *et al*, 2013), where the latter supports that p53 has not been inactivated by alternative mutations in other genes. Interestingly, the most recurrently deleted RPG in these cell lines was *RPL22* (Table EV4). Firstly, $Rpl22^{-/-}$ knockout mice have only subtle phenotypes with no significant translation defects, probably because these mice show increased expression of the paralog Rpl22-like1 (Rpl22 l1) which is incorporated in the ribosome instead of Rps22 (O'Leary *et al*, 2013). Secondly, *RPL22* has been identified as a potential tumor suppressor gene that is mutated or deleted in T-ALL and several epithelial tumor types (Rao *et al*, 2012; Novetsky *et al*, 2013; Goudarzi & Lindstrom, 2016). Another interesting observation was recurrent deletion of *RPS6*, which (like in the primary tumors) was always associated with co-deletion of *CDKN2A*. These data indicate that some RPG deletions are less likely to cause negative selection, either because of gene redundancy, because they do not activate p53, or because they are associated with a pro-proliferative effect that allows the cells to escape the negative effect of p53 activation.

### Further suppression of deleted RPGs inhibits the proliferation of cancer cells

The almost complete lack of homozygous RPG deletions in the three copy number data sets supports that such lesions are not tolerated. These observations are consistent with the notion that ribosomal proteins are essential for generation of functional ribosomes and cell survival. To investigate whether RPG haploinsufficiency renders cancer cells vulnerable to further suppression of ribosome function, we analyzed two sets of shRNA screening data from 102 and 72 genomically annotated cancer cell lines (Cheung *et al*, 2011; Marcotte *et al*, 2012). These pooled screens targeted 11,194 and 16,056 genes, including 26 and 55 RPGs, respectively. For each targeted gene, the effect of knockdown on proliferation in cell lines

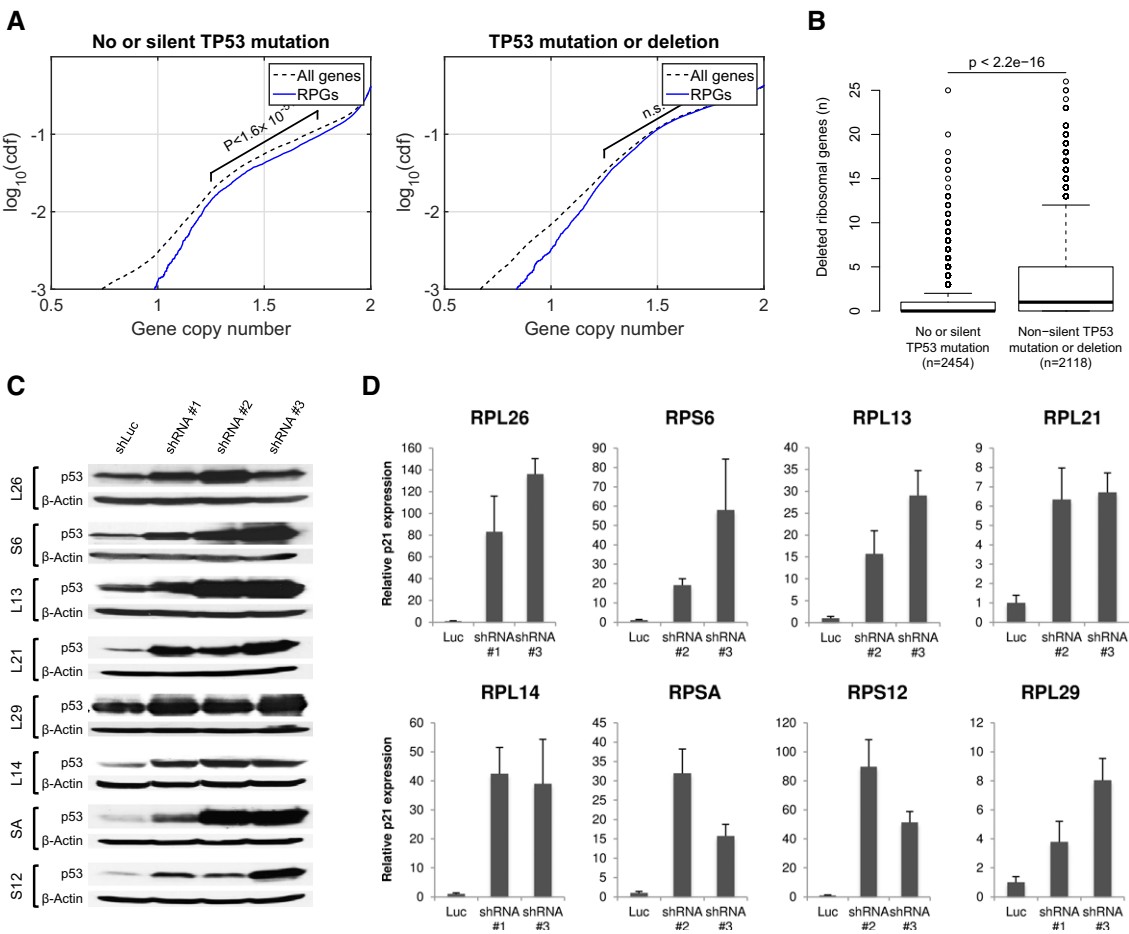

**Figure 2.  RPG haploinsufficiency associates with *TP53* mutation status and p53 activation.**

A    RPG deletions are underrepresented in *TP53*-intact tumors, but not in *TP53*-mutant tumors. Plots show cumulative copy number distributions for RPGs (blue) versus all genes (dashed black) for 4,675 TCGA samples having both copy number microarray and whole-exome sequence data. However, cases without *TP53* mutation/deletion show fewer copy numbers consistent with heterozygous loss of RPGs compared to other genes (brackets), whereas cases harboring *TP53* mutations/deletions show no corresponding difference. These plots also show that, regardless of *TP53* mutation status, homozygous loss of RPGs is extremely rare.

B    Number of hemizygous RPG deletions in *TP53*-mutant and -wild-type tumors. Boxes indicate medians and the first and third quartiles. Whiskers indicate first and third quartiles ±1.5 times the interquartile range. Notches indicate confidence intervals around the median. *P*-values indicate significance by Wilcoxon rank-sum test.

C, D    To obtain further support for p53-negative selection, we used shRNA-mediated knockdown of RPGs in A549 cells, which are *TP53*-wild type. Panels show p53 protein levels as analyzed by Western blot (β-actin loading control), and *P21* transcript levels as analyzed by qPCR (*ACTB* normalization control) after 4–6 days of expression of two shRNAs per tested RPG. Error bars indicate standard deviation between triplicates. Throughout, we observed increased p53 levels and *P21* expression. These data support that RPG deletions lead to p53 activation and negative selection in cancer cells.

carrying one gene copy, relative to the effect in cell lines carrying at least two gene copies, was scored (Marcotte *et al*, 2012; Shao *et al*, 2013). In both data sets, we observed strongly significant enrichments of negative gene scores among RPGs ($P = 1.0 \times 10^{-8}$ and $P = 1.7 \times 10^{-4}$, respectively; Fig 3A and B), indicating that further suppression of RPGs preferentially inhibits the growth of RPG-haploinsufficient cells.

**Deletion of RPGs influences rRNA maturation in cancer cells**

Previously, *in vitro* studies have shown that knockdown of *RPS6* and other 40S RPGs impairs the processing of pre-ribosomal RNA species into mature rRNA in ribosomopathies(Narla & Ebert, 2010; O'Donohue *et al*, 2010; Raiser *et al*, 2014). To explore whether RPG

haploinsufficiency leads to rRNA maturation defects also in cancer cells, we decided to analyze rRNA maturation across pediatric acute lymphoblastic leukemia (ALL) samples harboring CDKN2A (9p) deletions with and without concurrent deletion of *RPS6*. Deletions targeting *CDKN2A* are present in 20–30% of B-cell ALL and 95% of T-cell ALL. We first genotyped 47 previously banked pediatric ALL samples using copy number microarrays. For two samples (one with homozygous deletion of *CDKN2A* but intact *RPS6* and one with homozygous deletion of *CDKN2A* and hemizygous deletion of *RPS6*), we were able to obtain viable cells, which were xenografted into NOD *scid* gamma immunodeficient mice and expanded *in vivo*. RNA from sorted tumor cells was assessed by Northern blot to analyze pre-ribosomal RNA processing. Interestingly, the *RPS6*-haploinsufficient case exhibited rRNA maturation defects similar to

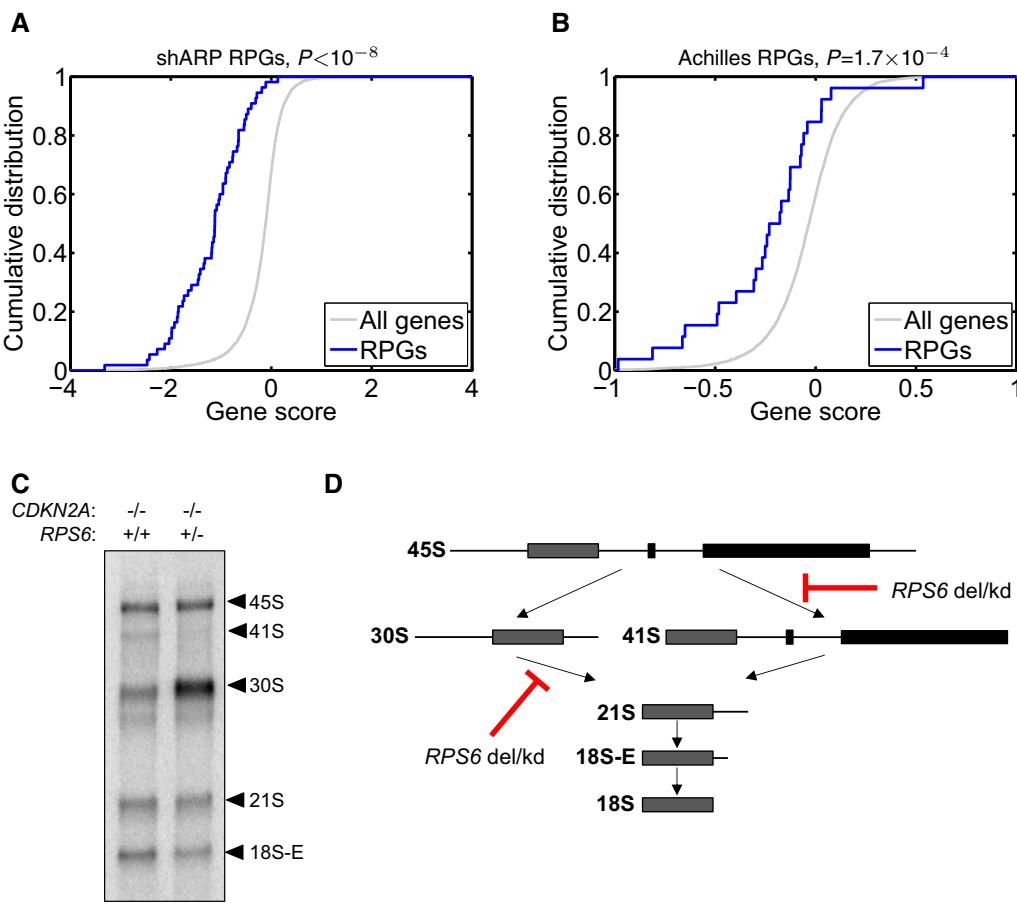

**Figure 3. Consequences of hemizygous RPG deletions.**

A, B To investigate whether RPG haploinsufficiency renders cancer cells susceptible to further suppression of ribosome function, we analyzed genomewide pooled shRNA screening data from the (A) shARP and (B) Achilles studies (data for 72 and 102 genomically annotated cancer cell lines, respectively). For each targeted gene, the effect of knockdown on proliferation in cell lines carrying one gene copy, relative to those carrying at least two gene copies, was scored. We observed strongly significant enrichments of negative gene scores for RPGs (solid blue; $P = 1.0 \times 10^{-8}$ and $P = 1.7 \times 10^{-4}$, respectively) compared to other genes (gray), demonstrating that further suppression of hemizygously deleted RPGs inhibits cell growth.

C, D (C) To explore whether RPG haploinsufficiency causes rRNA maturation defects in cancer cells, we focused on pediatric ALL with 9p deletions targeting *CDKN2A*. From two previously banked samples (one with homozygous *CDKN2A* deletion but intact *RPS6* and one with homozygous deletion of *CDKN2A* and hemizygous deletion of *RPS6*), we obtained viable cells, which were xenografted into NOD *scid* gamma immunodeficient mice and expanded *in vivo*. RNA from sorted tumor cells was assessed by Northern blot to analyze pre-ribosomal RNA processing. The RPS6-haploinsufficient case exhibited rRNA maturation defects similar to those seen previously in shRNA-targeted RPS6-deficient cells, namely (D) a reduction in 41S species (18S-E, 21S and 41S) and an accumulation of 30S species (30S and 45S). These observations provide the first evidence indicating that the rRNA maturation defect that is observed in ribosomopathies may also be present in RPG-haploinsufficient cancer cells.

those seen previously in shRNA-targeted *RPG*-deficient cells (cf. Narla & Ebert, 2010; Raiser *et al*, 2014 and references therein), namely a reduction in 41S species (18S-E, 21S, and 41S) and an accumulation of 30S species (30S and 45S), indicating that heterozygous deletion of *RPS6* impacts on ribosomal RNA biogenesis in ALL cells (Fig 3C and D). The difference in rRNA pattern observed in ALL cells with and without deletion of RPS6 was comparable to the difference in rRNA pattern observed in *TP53*-wild-type MOLM13 leukemia cells with and without shRNA knockdown of RPS6 (Fig EV7). Collectively, these observations provide the first evidence that the rRNA maturation defect that is observed in ribosomopathies on the basis of heterozygous inactivating mutations in other RPGs (Narla & Ebert, 2010; Raiser *et al*, 2014) may also be present in cancer cells with hemizygous RPG deletions.

## Discussion

Here we report the novel finding that ribosomal protein genes are routinely deleted across human cancers, particularly in concert with *TP53* mutation. Such a finding could lead to new possibilities for cancer therapy in *TP53*-mutant patients. Previously, mutation of RPGs (without concurrent *TP53* mutation) has primarily been associated with rare ribosomopathies, specific tumor subtypes, and cancer development in zebrafish models.

The analyses in this study are based on a large number of samples from primary samples, belonging to a broad range of different cancer types. The results indicate that RPG deletions are enriched among the samples that have concurrent *TP53* mutation. This finding is in accordance with previous studies on ribosomopathies (particularly

DBA and the 5q-syndrome) showing that RPG haploinsufficiency leads to activation of the p53 pathway, most likely through the 5S RNP-MDM2 pathway. One of the current main hypotheses is that the ribosomal assembly intermediate 5S ribonucleoprotein particle, containing RPL5, RPL11, and the 5S rRNA, accumulates when ribosome biogenesis is blocked; the excess 5S RNP binds to murine double minute 2 (MDM2), the main p53-suppressor in the cell, inhibiting its function and leading to p53 activation. Our data indicate that targeted degradation of the mRNAs for RPGs that are frequently deleted in cancer leads to p53 activation in A549 and MOLM13 cells, implicating p53 activation and (secondary to that) negative selection as a probable cause of the lower frequencies of RPG deletions seen among *TP53*-wild-type tumors. Moving forward, follow-up studies will be needed to examine whether this activation is mediated through the 5S RNP-MDM2 pathway. Additionally, analyses of clonal heterogeneity (e.g., using single-cell experiments or mathematical modeling of allele ratios) could illuminate whether p53 mutation precedes RPG deletion during oncogenesis.

Further, a central question about the biochemical consequences of RPG haploinsufficiency will be to determine precisely how these alterations affect rRNA maturation in cancer cells. Our data, though limited in sample size, indicate that rRNA patterns in acute lymphoblastic leukemia cells harboring *RPS6* deletion are perturbed in the same way as in cell lines with sh*RPS6* knockdown. These observations motivate further studies with larger numbers of samples to determine how variation in rRNA patterns associates with genetics lesions in RPGs.

Finally, genes that are hemizygously deleted are not necessarily drivers but can confer susceptibility for therapies (Muller *et al*, 2012; Nijhawan *et al*, 2012; Liu *et al*, 2015). Our study represents the first detailed examination of vulnerabilities in a specific cellular component. Our data show that RPG haploinsufficiency is a strikingly common vulnerability that is enriched among p53-deficient tumors, which are often hard to treat. Our results raise the question whether it could be possible to exploit RPG haploinsufficiency to selectively kill cancer cells. A number of drugs that modulate ribosomal function exist, including rapamycin and other inhibitors ("rapalogs") of the mammalian target of rapamycin (mTOR; Easton & Houghton, 2006), and compounds that inhibit ribosome biogenesis by inhibiting rRNA synthesis (Drygin *et al*, 2011; Bywater *et al*, 2012; Peltonen *et al*, 2014; Colis *et al*, 2014). These drugs are active against subsets of human tumors, but their therapeutic scope is unknown and could depend on the presence of ribosome defects. Future studies will inform how RPG deletions can be used for the treatment of a large subset of human cancer.

# Materials and Methods

### Cancer genome data sets

Firstly, we obtained normalized, segmented Affymetrix and Agilent DNA copy number data (downloadable.seg files) representing 7,225 primary tumor samples belonging to 24 tumor types from TCGA and with matched whole-exome sequencing data (somatic mutations; 4,675 samples) and RNA-sequencing data (4,919 samples; Cancer Genome Atlas Research Network *et al*, 2013; http://cancergenome. nih.gov/). Gene-wise copy numbers by averaging the copy number

signal across genes (from annotated gene start to gene end) in each sample. Secondly, we obtained corresponding data Affymetrix DNA copy number data representing 2,476 primary tumor samples belonging to 13 tumor types from Tumorscape (Beroukhim *et al*, 2010; http://www.broadinstitute.org/tumorscape), and for 1,043 cancer cell lines from the CCLE (Barretina *et al*, 2012; http://www.broad institute.org/ccle). To determine whether a gene was deleted, we applied $\log_2$ copy number thresholds between $-0.3$ and $-0.7$, corresponding to linear scale copy numbers 1.6 and 1.2, respectively. A $\log_2$ ratio of $-0.7$ corresponds to the theoretical copy number amplitude for a hemizygous deletion that present in all tumor cells in a sample with 80% tumor cell fraction (a TCGA inclusion criterion). This threshold is conservative as it does not leave any room for clonal heterogeneity, technical underestimation, or noise. The $\log_2$ ratio $-0.3$ corresponds to the midpoint between the theoretical copy number and the euploid, and is less conservative. To call homozygous deletions, we used a $\log_2$ ratio threshold of $-2.0$, corresponding to copy number 0.5 in linear scale. For completeness, we repeated our analyses with other reasonable thresholds, yielding results in broad agreement with those shown. The file processing and statistical analyses were done with Matlab (http://www.math works.com), R (http://www.r-project.org), Ultrasome (Nilsson *et al*, 2009), RenderCat (Nilsson *et al*, 2007), and various scripts. The set of RPGs was defined by the 78 annotated genes encoding the known proteins of the small and large ribosomal subunits (Fig 1).

### Lentiviral shRNA vectors and infection

Lentiviral shRNAs targeting frequently RPGs in the pLKO.1 or pLKO_TRC005 vector were obtained from the The RNAi Consortium (TRC) at the Broad Institute (Table EV3). Lentivirus was produced in 293TL cells. A549 cells were infected with 1 day after plating in the presence of 8 μg/ml polybrene (Sigma-Aldrich) and selected 24 h later with 2 μg/ml puromycin (Sigma-Aldrich) for at least 48 h before collection and processing for protein lysates or mRNA. A549 cells were maintained at 37°C and 5% $CO_2$ in F-12K medium (ATCC) supplemented with 10% fetal bovine serum and 1% pen/ strep/glutamine (Gibco). MOLM13 cells were transduced by spinfection with 8 μg/ml polybrene (Sigma-Aldrich). The cells were selected at 24 h after transduction with 0.3 μg/ml puromycin for 48 h. Puromycin-selected cells were harvested and divided as per experimental need for processing RNA or protein lysate. MOLM13 cells were maintained in RPMI 1640 supplemented with 10% fetal bovine serum and 1% PEST (Gibco).

### Western blot

Cells for protein analysis were lysed with Pierce® IP Lysis Buffer (Thermo Scientific). Western blots were performed using 25–50 μg of protein and primary antibodies against p53 (DO-1; Santa Cruz Biotechnology) at 1:500 dilution and β-actin (C4; Santa Cruz Biotechnology) at 1:3,000 dilution. HRP-conjugated anti-mouse secondary antibody (GE Healthcare) was used at 1:10,000 dilution. Immunoreactive proteins were visualized using SuperSignal® West Pico Chemiluminescence Substrate (Thermo Scientific). MOLM13 cells were lysed with Biorupter Pico (Diagenode) using 30/30 s on/ off for 10-min program in Laemmli sample buffer (Bio-Rad). Blots were probed with primary antibody against p53 (Sc-126, DO-1,

Santa Cruz Biotechnology) at 1:1,000 dilution and β-actin (Sc-8432, C-1; Santa Cruz Biotechnology) at 1:3,000 dilution.

## Quantitative real-time PCR

RNA was purified from A549 and MOLM13 cells using TRIzol (Invitrogen) and RNeasy mini kit (cat no 74104), respectively. First-strand cDNA was generated using 100–200 ng of total RNA and oligo(dT) primers with the Superscript III reverse transcription kit (Invitrogen), and Omniscript RT kit (cat no 205113) was used for cDNA synthesis from MOLM13 cells. Quantitative RT–PCR was performed using TaqMan® Gene Expression Master Mix (Applied Biosystems) or SYBR® Green PCR Master Mix (Applied Biosystems) on an ABI Prism 7900HT system and StepOnePlus Real-Time PCR system (Applied Biosystems). We used TaqMan® assays to assess the knockdown effect for *RPS6* (Hs01058685_g1), *RPL26* (Hs00864008_m1), *RPL13* (Hs00744303_s1), *RPL21* (Hs03003806_g1), *RPL29* (Hs00426490_g1), *RPL14* (Hs03004339_g1), *RPSA* (Hs00347791_s1), and *RPS12* (Hs04184906_g1) using *ACTB* (ABI #401846) as control, and *P21* (Hs00355782_m1), *NOXA* (Hs00560402_m1), PUMA (Hs00248075_m1), *BAX* (Hs00180269_m1) were assessed using *GAPDH* (Hs02758991_g1) as endogenous control for MOLM13 cells. Quantitative PCR for *P21* and *ACTB* was performed using SYBR® Green PCR Master Mix (Applied Biosystems) and the following primers: *P21* forward 5′-GCTCTGCTGCAGGGGACAGC-3′; *P21* reverse 5′-GCCGCCGTTTTCGACCCTGA-3′; *ACTB* forward: 5′-AGCGAGCATCCCCCAAAGTT-3′; and *ACTB* forward: 5′-GGGCAC-GAAGGCTCATCATT-3′ for A549 cells.

## Analysis of leukemia samples

We obtained DNA extracted from 47 blood and bone marrow samples taken at diagnosis from patients with pediatric ALL. The samples were collected and banked at the Dane Farber Cancer Institute subject to informed consent (Dana Farber Cancer Institute, institutional review board protocol no. 05-001). The experiments conformed to the principles set out in the WMA Declaration of Helsinki [http://www.wma.net/en/30publications/10policies/b3/] and the NIH Belmont Report [https://www.hhs.gov/ohrp/regulations-and-policy/belmont-report/]. The samples were analyzed on Affymetrix 6.0 arrays (Affymetrix Inc.; data available for download from the ArrayExpress repository accession no. E-MTAB-5450). Copy number changes were delineated using the program Ultrasome (Nilsson *et al*, 2009). We obtained banked tumor cells and expanded them *in vivo* in NOD *scid* gamma immunodeficient mice for about 6 months. When the mice appeared moribund, they were sacrificed. Cells harvested from the spleens and bone marrow were stained with anti-human CD45 (Miltenyi Biotec) and sorted by fluorescence-activated cell sorting to purify tumor cells, from which RNA was isolated for rRNA maturation analysis by Northern blot (Qiagen kits). For Northern blot analysis, gel-fractionated RNA was transferred to zeta-probe membranes (Bio-Rad). An oligonucleotide probe 5′-CCTCGCCCTCCGGGCTCCGTTAATGATC-3′ (complementary to sequences 5520–5547 spanning the boundary between 18S rRNA and ITS1) was labeled with $^{32}$P using T4 polynucleotide kinase and hybridized overnight with membrane-bound RNA at 37°C in ULTRAHyb-Oligonucleotide hybridization buffer (Ambion). Membranes were washed at 37°C with 6× SSC and subjected to

phosphorimager analysis. MOLM13 RPS6 knockdown RNA samples were electrophoretically fractioned in MOPS-gel running buffer (cat no AM8671, Ambion) and transferred to Biodyne nylon membrane (cat no 77016, Thermo Scientific) using semi-dry blot method. Hybridization and washing steps were followed as recommended in Northern Max kit (AM1940, Thermo Scientific). The biotinylated cDNA probe hybridization was visualized following Chemiluminescent Nucleic Acid Detection Module (cat no 89880, Thermo Scientific).

## The paper explained

### Problem

The human ribosome consists of an rRNA scaffold and about 80 proteins, divided into two subunits. Ribosomal protein genes (RPGs) are denoted *RPS1*, *RPS2*, etc. for the small (40S) subunit; *RPL1*, *RPL2*, etc. for the large (60S) subunit. Increasing evidence holds that mutation of RPGs leads to specific clinical and cellular phenotypes, including Diamond-Blackfan anemia, the 5q- subtype of myelodysplastic syndrome, and specific tumor types. Furthermore, many RPGs are tumor suppressor genes in animal models_ENREF_13. Despite these observations, which support a link between RPG lesions and cancer, the occurrence of RPG lesions in human cancers has not been investigated systematically.

### Results

We carried out a large-scale analysis of cancer genome data to determine the frequency and selective pressure of RPG lesions across human cancers. We first looked for chromosomal deletions and amplifications and point mutations in RPGs using pre-existing DNA copy number microarray and whole-exome sequencing data from a total of 10,744 cancer specimens and cell lines. Because recent studies have shown that RPG haploinsufficiency activates p53 in ribosomopathies, and the pathobiology of ribosomopathies can be alleviated by p53 inhibition, we hypothesized inactivation of RPGs could lead to negative selection unless the cells have mutated *TP53*, and, accordingly, looked for associations between inactivating RPG lesions and *TP53* mutation. While we observed few point mutations and homozygous deletions in RPGs, we detected hemizygous RPG deletions in about 43% of samples. Consistent with negative selection, further analyses revealed an underrepresentation of RPG deletions in *TP53*-intact tumors, whereas we did not see any signs of negative selection in *TP53*-mutant tumors. Furthermore, functional experiments showed that deficiency of frequently deleted RPGs increases p53 activity in *TP53*-intact cell lines and perturbs rRNA maturation both in cell lines cultured *ex vivo* and in primary acute leukemia cells with specific RPG deletions and expanded *in vivo* in xenograft models. Finally, consistent with the low frequency of homozygous deletion, analysis of genome-wide shRNA screening data showed that further suppression of hemizygously deleted RPGs inhibits the growth of RPG-haploinsufficient cancer cells.

### Impact

Genes that are hemizygously deleted are not necessarily drivers but can confer susceptibility for therapies. Our data show that RPG haploinsufficiency is a strikingly common vulnerability that is enriched among p53-deficient tumors, which are often hard to treat. Our results raise the question whether it could be possible to exploit RPG haploinsufficiency to selectively kill cancer cells. A number of drugs that modulate ribosomal function exist. These drugs are active against subsets of human tumors, but their therapeutic scope is unknown and could depend on the presence of ribosome defects. Future studies will inform how RPG deletions can be used for the treatment of a large subset of human cancer.

**Analysis of shRNA screening data**

To test whether further suppression of hemizygously deleted RPGs inhibits cell growth, we used genomewide, pooled shRNA screening data from the Achilles and shARP studies (Cheung *et al*, 2011; Marcotte *et al*, 2012). The Achilles study provided data on 102 cell lines infected with a pool of lentivirally delivered shRNAs, composed of 54,020 shRNAs targeting 11,194 genes. The shARP study provided data on 72 cancer cell lines infected with a pool of 78,432 shRNAs targeting 16,056 genes. Gene scores reflecting the effect on knockdown on cell growth in cell lines carrying one copy versus at least two copies were provided with the original publications (Cheung *et al*, 2011; Marcotte *et al*, 2012; Shao *et al*, 2013). To test for enrichment of negative scores (indicating depletion) among RPGs compared to other genes, we used RenderCat (Nilsson *et al*, 2007).

**Expanded View** for this article is available online.

## Acknowledgements

The project was supported by research grants from the Swedish Foundation for Strategic Research (ICA08-0057), the Marianne and Marcus Wallenberg Foundation (2010.0112), the Swedish Children's Cancer Fund (PR2012-0084), and a Wallenberg Academy Fellows Award to B.N. (2012.0193). The project was also supported by the National Institute of Health (R01 HL082945 and P01 CA108631) and a Leukemia and Lymphoma Society Scholar Award to B.L.E.

## Author contributions

BN and BLE designed and supervised the project, performed computational analyses, and wrote the manuscript with input from DMW, SA, and SRE. Experiments were done by DR, RA, MM, BB and BM. Additional computational analyses were done by MJ and GS. All authors contributed to the final manuscript.

## Conflict of interest

The authors declare that they have no conflict of interest.

## For more information

The Cancer Genome Atlas (http://cancergenome.nih.gov/)
Tumorscape (http://www.broadinstitute.org/tumorscape)
Cancer Cell Line Encyclopedia (http://www.broadinstitute.org/ccle)
Ultrasome (http://www.broadinstitute.org/ultrasome)
Matlab (http://www.mathworks.com)
R project (http://www.r-project.org)

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
