## [Review Process File · EMBO Molecular Medicine]

Deletion of ribosomal protein genes is a common vulnerability in human cancer, especially in concert with TP53 mutations

Ram Ajore, David Raiser, Marie McConkey, Magnus Jöud, Bernd Boidol, Brenton Mar, Gordon Saksena, David M. Weinstock, Scott Armstrong, Steven R. Ellis, Benjamin L. Ebert, Björn Nilsson

*Corresponding authors: Benjamin L. Ebert, Broad Institute
Björn Nilsson, Lund University*

Review timeline:

Submission date:	02 June 2016
Editorial Decision:	07 July 2016
Revision received:	22 December 2016
Editorial Decision:	18 January 2017
Revision received:	25 January 2017
Accepted:	26 January 2017

Transaction Report:

Editor: Roberto Buccione

1st Editorial Decision

07 July 2016

Thank you for the submission of your manuscript to EMBO Molecular Medicine. We have now heard back from the Reviewers whom we asked to evaluate your manuscript.

We are sorry that it has taken longer than usual to get back to you on your manuscript. In this case we experienced difficulties in securing appropriate reviewers and then obtaining their evaluations in a timely manner. Further to this, I wished to discuss the evaluations further with my colleagues

As you will see, all three reviewers are quite positive, although Reviewers 1 and 3 appear more reserved.

In aggregate, significant concerns were raised including 1) use of a single cell line, 2) insufficient analysis of the role of p53, 3) unclear causal connection between rRNA processing and S6 mutation, 4) insufficient information to support reproducibility (a topic very close to our hearts here at EMBO) and other issues.

After reviewer cross-commenting and further discussion, it was acknowledged that although much of the manuscript is based on association, improvements are nevertheless needed to tighten the overall significance. For instance the knock down experiments to show TP53/p21

induction should be improved, in particular leveraging the CCE genomics analysis, which should allow better qualification of which ribosomal genes are or are not deleted in the model lines (TP53 mutated, TP53 WT).

It was agreed that you should be allowed to submit a revised manuscript, with the understanding that the Reviewers' concerns must be addressed with additional experimental data where appropriate and that acceptance of the manuscript will entail a second round of review.

It is important that you consider that it is EMBO Molecular Medicine policy to allow a single round of revision only and that, therefore, acceptance or rejection of the manuscript will depend on the completeness of your responses included in the next, final version of the manuscript.

I look forward to seeing a revised form of your manuscript as soon as possible.

***** Reviewer's comments *****

Referee #1 (Comments on Novelty/Model System):

Single cell line model (A549 lung cancer line)

Referee #1 (Remarks):

The MS combines (1) comprehensive bioinformatic analysis of public sequencing and copy number data to define statistical association between loss of function mutations involving ribosomal protein genes (RPG) in cancer and TP53 status (2) in vitro knockdown of selected RPG in the TP53 WT lung cancer line A549 to show altered expression of RPG enhances p53 protein levels and p21 transcription (3) bioinformatic analysis of shRNA screening data show association between enrichment of RPG in lethality and (4) analysis of RNA maturation in primary acute lymphocytic leukaemia cells to show that rRNA maturation defects are associated with loss of RPS6 in cells with CDKN2A (9p) deletion.

Major comments

1. There are no detailed bioinformatic methods that would allow reproducible analysis or in depth understanding by this reviewer. The MS would be greatly strengthened by at least detailed experimental plan showing assumptions made on gene data and software used. Please consider weave or Rmd document for major findings using (if needed) intermediate data objects that are also published with the MS.

Minor comments only

1. Page 4. Instead of using the second page of MS to describe the results of MS, please widen explanation of aims and approaches, as this will be more accessible for the general reader (and won't duplicate the abstract).
2. P values should be quoted as $\ll 0.001$ where relevant.

Referee #2 (Remarks):

The Manuscript by B. Nilsson et al. addresses the relevance of Ribosomal protein genes' (RPG) deletion in human cancer, and their association with p53 status. Germline heterozygous inactivating mutations/deletions of some RPGs cause Diamond-Blackfan anemia and disorders known as ribosomopathies, while acquired somatic mutations are associated to some human malignancies.

In search of potential vulnerabilities of mutant p53 cancers, the authors hypothesized that p53 mutation/inactivation suppresses the vulnerability of RPG haploinsufficient cancer cells, and that

this may represent a potential therapeutic target.

The topic is timely and of potential broad interest in the field of cancer and cell biology. The authors provide several original observations in support of their hypotheses, however the results are largely based on correlative observations. Their conclusions appear to be premature, and more experimental work would be required to support their conclusions and to assess the relevance of their findings, before the manuscript is considered for publication.

Major comments:

- Through bioinformatics analyses of large human cancer datasets, the authors unveiled the presence of hemizygous deletions of many RPGs in a large proportion of human cancers, inversely correlating with p53 function. This suggests a potential general role of RPGs mutations in the pathogenesis of cancer. The knockdown of several RPGs they found altered in human cancers, induced p53 protein and transcription of the p53 target P21, in lung adenocarcinoma A549 cell line. These findings would be better supported by extending the analysis to additional wild-type p53 harboring cancer cell lines, and upon silencing RPGs in the isogenic mutant p53 context. Moreover, silencing RPGs in p53-silenced A549 cells with overexpression of mutant p53 variant(s) might further underline the specific role of RPGs depletion in wild-type p53 activation. Importantly, little evidence of induction of p53 functions is provided beyond P21 expression. For example, additional wild-type p53 targets (e.g. PUMA/NOXA) could be tested to strengthen the claim of activation of the wild-type p53 transcriptional activity. The effects of p53 induction should also be directly tested;

- To assess whether RPG haploinsufficiency might render cancer cells vulnerable to further suppression of ribosome function, the authors analyzed the effect of RPG dosage on cell proliferation upon RPG knockdown, in several genomically annotated cancer cell lines, using genome-wide screen data from shARP and Achilles studies. They found indications that further suppression of hemizygous RPG expression is associated to cell growth inhibition. It would be interesting to know the p53 status of those cell lines;

- The authors show that xenografted CDKN2A-deficient RPS6 haploinsufficient ALL cells exhibit defects in ribosomal RNA maturation, similarly to what was previously observed in ribosomopathies. To test the authors' hypothesis, it would be interesting to assess whether further reduction of RPG function elicits cancer cells vulnerability in this model. In addition, to assess the relevance of altered rRNA maturation, reconstitution experiments might be performed to examine the effect of normalized levels of those rRNA that are affected by deletion of their gene on tumor cell features.

Referee #3 (Comments on Novelty/Model System):

I have a bit of issue with the technical approach to Figure 3C,D which I have detailed below.

Referee #3 (Remarks):

This is a nicely written and straightforward paper that reveals an important and novel finding about how deletions in RP genes are linked to p53 inactivation and cancer. I believe it will be very worthy of publication in EMBO Molecular Medicine if the following concerns are addressed:

Major comments:

1. I am a bit wary of the rRNA northern blot experiment comparing the pediatric ALL cells with or without RPS6 deletions (Figure 3C,D). Since these cancer cells were derived from presumably unrelated patients, I don't think it may be state definitively that the changes in rRNA processing are due to the presence of the S6 mutation since one cannot exclude the possibility that there are other mutations in the cells with the S6 mutation also affecting ribosome biogenesis.

To control for these potential background mutations, perhaps the better experiment would be to do the add back experiment and exogenously re-express the RP that is deleted in either the same

primary cells or in a cancer cell line with a known RP mutation (for example EOL-1 cells carry a mutation in RPS6, NB-4 cells carry an RPL14 deletion...).

Alternately, what might be more convincing would be to show 4-5 controls (with WT RPs) instead of only 1. If all the controls display the same rRNA processing pattern, at least the authors have a better case.

Lastly, I think that this experiment would be really benefit from a control knocking down RPS6 and showing that in this specific cell type the skewing of the rRNA looks the same as with the mutation. No citations are mentioned here referring to the previous experiments measuring rRNA processing where the authors claim that their results look similar to what has been previously shown:

"Interestingly, the RPS6-haploinsufficient case exhibited rRNA maturation defects similar to those seen previously in shRNA-targeted RPS6-deficient cells, namely a reduction in 41S species. The Narla and Raiser papers cited at the end of the paragraph do not present northern blots. Upon searching the literature I found Robledo et al (RNA, 2008) where in Figure 5 they examine rRNA processing in HeLa cells with knocked down RPS6 and these results look different than what the authors present in Figure 3C (the knockdown of RPS6 in HeLa cells show a substantial loss of 21S which I don't see in the CDKN2A^{-/-};RPS6^{+/-} cells).

Minor suggestions:

1. Two papers by Amsterdam et al and MacInnes et al are cited in the second paragraph of the introduction, "Eleven RPGs are tumor suppressor genes in zebrafish, where hemizygous inactivation of these genes cause malignant peripheral nerve sheath tumors with high penetrance". However, these reports concluded that in none of the tumors with RPG mutations was p53 found genetically inactivated. This is also the case with the T-ALL cancers reported by De Keersmaecker et al. Some discussion of this I think would be interesting, for it does appear to be the case at least with the zebrafish tumors that the cells have found an alternate mechanism to reduce p53.

2. On page 6, the statement is made "In DBA and the 5q- syndrome, RPG haploinsufficiency perturbs the stoichiometry of ribosomal proteins, leading to inefficient ribosome assembly and increased concentrations of free ribosomal proteins, some of which (e.g., RPL11) bind MDM2 and inhibit its ability to target p53 for proteasomal degradation." To the best of my knowledge it has not ever been shown experimentally that there is increased binding of MDM2 to RPL11 (or any other RP) in the presence of a bona fide DBA-linked RP gene mutation...most of these reports (including those cited here) are using actinomycin D to disrupt ribosome biogenesis. It's a nice hypothesis, but in the absence of experimental data should not be stated as fact.

1st Revision - authors' response

22 December 2016

Response to comments

We thank the three referees and the editor for their constructive comments. All points raised have been addressed. A systematic list of the changes made is enclosed below. The changes have also been indicated in the revised manuscript.

Referee #1:

"Single cell line model (A549 lung cancer line)."

DONE. We agree. We have now repeated the knockdown experiments in a second *TP53* wildtype cell line (MOLM13) for three frequently deleted RPGs (*RPS6*, *RPL13*, and *RPL26*). As shown in the new **Supplementary Figures 5 and 6**, we observed increased p53 protein levels and *P21* transcript levels, just like in A549 cells. It is also important to point out that the relationship between RPG haploinsufficiency and p53 activation has been demonstrated for other RPGs in several studies, some of which are referred to in the manuscript.

“There are no detailed bioinformatic methods that would allow reproducible analysis or in depth understanding by this reviewer.”

DONE. Additional details have been added in the Method section (pages 13 to 17). We have also added web links or references to all data and software. The results can be reproduced based on the provided information by bioinformaticians skilled in cancer genomics.

“Page 4. Instead of using the second page of manuscript to describe the results, please widen explanation of aims and approaches, as this will be more accessible for the general reader.”

DONE. We thank for the reviewer for this suggestion. The section on page 4 has now been improved as requested.

“P values should be quoted as $\ll 0.001$ where relevant.”

DONE. Very small P values now quoted as $\ll 10^{-10}$.

Referee #2:

“The topic is timely and of potential broad interest in the field of cancer and cell biology. The authors provide several original observations in support of their hypotheses.”

THANKS. We thank the reviewer for his/her encouragement.

“The findings would be better supported by extending the analysis to additional wild-type p53 harboring cancer cell lines”.

DONE. See corresponding comment from Referee #1 above.

“Little evidence of induction of p53 functions is provided beyond P21 expression. For example, additional wild-type p53 targets (e.g. PUMA/NOXA) could be tested to strengthen the claim of activation of the wild-type p53 transcriptional activity.”

DONE. We thank the reviewer for this excellent suggestion. Towards this, we measured the expression of additional p53 target genes in the knock-down experiments in MOLM13 cells. As shown in the new **Supplementary Figure 6**, the expression of *BAX*, *PUMA*, and *NOXA* increases in the same manner as *P21*, further supporting a functional activation of p53. Additionally, previous studies on ribosomopathies have shown that deficiency of other RPGs, particularly RPS14 and RPS19, leads to increased p53 activity, and that the pathobiology of ribosomopathies can be alleviated by p53 inhibition (references given in the manuscript). Our data support that the additional RPGs identified in this study activate p53 just like the others.

“It would be interesting to know the p53 status of the shARP and Achilles cell lines.”

We agree. Unfortunately, we could not obtain this information.

“The authors show that xenografted CDKN2A-deficient RPS6 haploinsufficient ALL cells exhibit defects in ribosomal RNA maturation, similarly to what was previously observed in ribosomopathies. To test the authors' hypothesis, it would be interesting to assess whether further reduction of RPG function elicits cancer cells vulnerability in this model. In addition, to assess the relevance of altered rRNA maturation, reconstitution experiments might be performed to examine the effect of normalized levels of those rRNA that are affected by deletion of their gene on tumor cell

features.”

We agree that this would be interesting. Unfortunately, these cells do not serially transplant, and we were therefore too limited in cellular material to carry out these experiments.

Referee #3:

“This is a nicely written and straightforward paper that reveals an important and novel finding about how deletions in RP genes are linked to p53 inactivation and cancer. I believe it will be very worthy of publication in EMBO Molecular Medicine”

THANKS. We thank the reviewers for his/her encouragement.

“I have a bit of issue with the technical approach to Figure 3C,D”, “I am a bit wary of the rRNA northern blot experiment comparing the pediatric ALL cells with or without RPS6 deletions (Figure 3C,D). Since these cancer cells were derived from presumably unrelated patients, I don't think it may be state definitively that the changes in rRNA processing are due to the presence of the S6 mutation since one cannot exclude the possibility that there are other mutations in the cells with the S6 mutation also affecting ribosome biogenesis.”, “I think that this experiment would be really benefit from a control knocking down RPS6 and showing that in this specific cell type the skewing of the rRNA looks the same as with the mutation.”

DONE. This is an excellent point. To address this, we recorded rRNA maturation patterns in *RPS6* wildtype vs knockdown MOLM13 leukemia cells by Northern blot. As shown in the new **Supplementary Figure 7**, the differences in rRNA patterns of these cells were comparable to those of ALL cells with vs without *RPS6* deletion shown in **Figure 3C, D**.

“No citations are mentioned here referring to the previous experiments measuring rRNA processing where the authors claim that their results look similar to what has been previously shown: “Interestingly, the RPS6-haploinsufficient case exhibited rRNA maturation defects similar to those seen previously in shRNA-targeted RPS6-deficient cells, namely a reduction in 41S species. The Narla and Raiser papers cited at the end of the paragraph do not present northern blots. Upon searching the literature I found Robledo et al (RNA, 2008) where in Figure 5 they examine rRNA processing in HeLa cells with knocked down RPS6 and these results look different than what the authors present in Figure 3C (the knockdown of RPS6 in HeLa cells show a substantial loss of 21S which I don't see in the CDKN2A-/-;RPS6+/-cells).”

DONE. We thank the reviewer for bringing our attention to this point. This comment is partly due to a typo (“RPS6-deficient” should be “RPG-deficient”), which has now been corrected (page 11). Further, as mentioned in the response to the previous comment, we carried out knock-down experiments with shRNAs towards *RPS6* to demonstrate a causal connection between RPS deletion and skewing of rRNA patterns. As shown in the new **Supplementary Figure 7**, these experiments support that *RPS6* deficiency causes the type of rRNA skewing shown in **Figure 3**. Finally, the skewing of rRNA patterns towards less mature forms has been associated with haploinsufficiency of other RPGs in studies on ribosomopathies (review references added; page 11).

“Two papers by Amsterdam et al and MacInnes et al are cited in the introduction. However, these reports concluded that in none of the tumors with RPG mutations was p53 found genetically inactivated. This is also the case with the T-ALL cancers reported by De Keersmaecker et al. Some discussion of this I think would be interesting, for it does appear to be the case at least with the zebrafish tumors that the cells have found an alternate mechanism to reduce p53.”

DONE. We agree that these are fascinating observations. This point has now been clarified in the Discussion section (page 11).

“On page 6, the statement is made “In DBA and the 5q-syndrome, RPG haploinsufficiency perturbs the stoichiometry of ribosomal proteins, leading to inefficient ribosome assembly and increased concentrations of free ribosomal proteins, some of which (e.g., RPL11) bind MDM2 and inhibit its ability to target p53 for proteasomal degradation.” To the best of my knowledge, it has not ever been shown experimentally that there is increased binding of MDM2 to RPL11 (or any other RP) in the presence of a bona fide DBA-linked RP gene mutation...most of these reports (including those cited here) are using actinomycin D to disrupt ribosome biogenesis. It's a nice hypothesis, but in the absence of experimental data should not be stated as fact.”

DONE. Wording adjusted. References added (page 6).

Response to comments from the editor

“As you will see, all three reviewers are quite positive.”

THANKS. We thank the editor and referees for their encouragement and constructive comments. We are happy to hear that our work was appreciated.

“Concerns were raised including 1) use of a single cell line, 2) insufficient analysis of the role of p53, 3) unclear causal connection between rRNA processing and S6 mutation, 4) insufficient information to support reproducibility (a topic very close to our hearts here at EMBO) and other issues.”, “improvements are needed to tighten the overall significance. For instance, the knock down experiments to show TP53/p21 induction should be improved, in particular leveraging the CCLE genomics analysis, which should allow better qualification of which ribosomal genes are or are not deleted in the model lines (TP53 mutated, TP53 WT).

DONE. These are all excellent points. Points 1 to 4 have been addressed as described above. Moreover, the editor's suggestion to make better use of CCLE turned out to be interesting. To address this point, we first integrated the different layers of CCLE genomics data to determine TP53 status (as described in the reference Sonkin et al 2013). We then used the drug response data from CCLE to selected cell lines that were responsive to Nutlin-3, a well-known p53 activating drug. By looking specifically at cell lines that respond to Nutlin-3, we could select cell lines that are unlikely to have inactivated p53 through alternative mutations in other genes. Through this analysis, we identified RPG deletions that are permissible in cancer cells with intact p53 functions. Interestingly, the most frequently deleted RPGs in these cell lines turned out to be RPL22 (which is a tumor suppressor in several human tumor types, and, at least in mice, has a paralog that can be incorporated into the ribosome in its place) and RPS6 (always co-deleted with the tumor suppressor CDKN2A). These data indicate that some RPG deletions are less likely to cause negative selection, either because of gene redundancy, because they do not activate p53, or because they are associated with a pro-proliferative effect that allows the cells to escape the negative effect of p53 activation. These results are described on 9 and in the new **Supplementary Table 4**.

“EMBO Molecular Medicine now requires a complete author checklist to be submitted with all revised manuscripts. Provision of the author checklist is mandatory at revision stage.”

DONE. Author check list added.

“We now mandate that all corresponding authors list an ORCID digital identifier.”

DONE. Digital identifier added.

“The paper explained: EMBO Molecular Medicine articles are accompanied by a summary of the articles to emphasize the major findings in the paper and their medical implications for the non-specialist reader. Please provide a draft summary of your article highlighting -the medical issue you are addressing, -the results obtained and -their clinical impact.”

DONE. Summary added.

“A Conflict of Interest statement should be provided in the main text”

DONE. Conflict of interest statement added.

“There is space at the end of each article to list relevant web links for further consultation by our readers. Could you identify some relevant ones and provide such information as well?”

DONE. Web links added.

“Author contributions: the contribution of every author must be detailed in a separate section (before the acknowledgments).”

DONE. Author contributions added.

“Every published paper now includes a 'Synopsis' to further enhance discoverability. Synopses are displayed on the journal webpage and are freely accessible to all readers.”

DONE. Synopsis added.

“Figure panels should be indicated by capital letters (A, B, C etc).”

DONE. Figure panels indicated by capital letters.

2nd Editorial Decision

18 January 2017

Thank you for the submission of your revised manuscript to EMBO Molecular Medicine. We have now received the enclosed reports from the referees that were asked to re-assess it. As you will see, while reviewer 2 is now satisfied, reviewer 1 has a few remaining points concerning the level experimental support for some of the conclusions. Unless you have additional data that would address such issues, I would ask you to carefully and fully address them, including by introducing cautionary statements on the relative limitations in your manuscript.

Depending on the completeness of your response, I am willing make an editorial decision on your manuscript. Please highlight your changes in the manuscript text in the next version.

In the event of a positive outcome, there are a number of editorial requirements for you to comply with before we can proceed with acceptance. I suggest you do so for your next, final revision to reduce manuscript back and forth with the editorial office. The requested amendments are as follows:

- 1) The "weblinks" section should be renamed "For more information".
- 2) The "The Paper Explained" section should be moved to the manuscript main text

3) All the supplementary figures should be combined into a single PDF file with a first page table of contents, as per our Author Guidelines (<http://embomolmed.embopress.org/authorguide#expandedview>). Please also consequently use the appropriate figure callouts in the manuscript.

4) Please provide Table 1 as a doc/xls file.

5) Data described in submitted manuscripts should be deposited in a MIAME-compliant format with one of the public databases. We would therefore ask you to submit your microarray data to the ArrayExpress database maintained by the European Bioinformatics Institute for example. ArrayExpress allows authors to submit their data to a confidential section of the database, where they can be put on hold until the time of publication of the corresponding manuscript. Please see <http://www.ebi.ac.uk/arrayexpress/Submissions/> or contact the support team at arrayexpress@ebi.ac.uk for further information.

6) For experiments involving human subjects the authors must identify the committee approving the experiments and include a statement that informed consent was obtained from all subjects and that the experiments conformed to the principles set out in the WMA Declaration of Helsinki [<http://www.wma.net/en/30publications/10policies/b3/>] and the NIH Belmont Report [<http://ohsr.od.nih.gov/guidelines/belmont.html>]. Any restrictions on the availability or on the use of human data or samples should be clearly specified in the manuscript. Any restrictions that may detract from the overall impact of a study or undermine its reproducibility will be taken into account in the editorial decision. In this case I refer specifically to the data obtained from the blood and bone marrow samples taken at diagnosis from patients with pediatric ALL. We note that this was not mentioned in the checklist either. If applicable the checklist should also be amended and the new version uploaded with the revised manuscript

7) We encourage the publication of source data, with the aim of making primary data more accessible and transparent to the reader. Would you be willing to provide a PDF file per figure that contains the original, uncropped and unprocessed scans of all or at least the key gels used in the manuscript and/or source data sets for relevant graphs? The files should be labeled with the appropriate figure/panel number, and in the case of gels, should have molecular weight markers; further annotation may be useful but is not essential. The files will be published online with the article as supplementary "Source Data" files. If you have any questions regarding this just contact me.

8) You are welcome to suggest a striking image or visual abstract to illustrate your article. If you do please provide a jpeg file 550 px-wide x 400-px high

Please submit your revised manuscript within two weeks. I look forward to seeing a revised form of your manuscript as soon as possible.

***** Reviewer's comments *****

Referee #2 (Remarks):

B. Nilsson et al. have provided a revised version of the manuscript "Deletion of ribosomal protein genes is a common vulnerability in human cancer, particularly in concert with TP53 mutation", addressing the relevance of deletion of Ribosomal protein genes (RPG) in human cancer, and its association with p53 status.

The authors added further data in favor of their conclusions, but I still have a couple of issues:
 - the analysis of p53 function in RPG knockdown experiments;
 - the causal relationship between rRNA processing and S6 mutation. While the availability of ALL cells might be limiting to perform rescue experiments, the authors did not exploit alternative strategies.

Both of these questions have remained only partially addressed. Overall, the manuscript makes an interesting case on the association between RPG haploinsufficiency and p53 mutation, still the authors' conclusions appear premature at this stage.

Referee #3 (Remarks):

I have no further comments. The manuscript is suitable for publication.

2nd Revision - authors' response

25 January 2017

We thank the referees and the editorial team for their constructive comments. All points have now been addressed. The changes are as follows:

Referee #2

“The authors added further data in favor of their conclusions, but I still have a couple of issues: -the analysis of p53 function in RPG knockdown experiments; -the causal relationship between rRNA processing and S6 mutation. While the availability of ALL cells might be limiting to perform rescue experiments, the authors did not exploit alternative strategies. Both of these questions have remained only partially addressed.”

DONE. We agree with the referee that our work raises several new questions. It would definitely be interesting to try to answer all of these. Yet, it would require an effort that is beyond the scope of this study. However, to address the reviewer's request to the extent we can, we have added a description of the limitations of our current study in the Discussion section (pages 11-12).

Referee #3

“I have no further comments. The manuscript is suitable for publication.”

THANKS. We thank the reviewer for his/her encouragement. We are happy to hear that our work is appreciated.

Response to comments from the editor

“While reviewer #3 is now satisfied, reviewer #2 has a few remaining points concerning the level experimental support for some of the conclusions. Unless you have additional data that would address such issues, I would ask you to carefully and fully address them, including by introducing cautionary statements on the relative limitations in your manuscript.”

DONE. As mentioned above, we have now added a description of the limitations of our study to the Discussion section (pages 11-12). The new text includes cautionary statements on the relative limitations of our study, as well as a description of new questions that can potentially be addressed in follow-up studies by us or others.

“The ‘weblinks’ section should be renamed ‘For more information’”

DONE. Section renamed.

“The ‘The Paper Explained’ section should be moved to the manuscript main text”

DONE. Section moved to the main text. As requested, it is placed after the Acknowledgements, and the headings have been renamed Background-Results-Impact.

“All the supplementary figures should be combined into a single PDF file”

DONE. All expanded view figures have now been combined into a single file. For completeness, we also provide individual files for each of the figures.

“Please also consequently use the appropriate figure callouts in the manuscript.”

DONE. Figure and table callouts corrected.

“Please provide Table 1 as a doc/xls file.”

DONE. Excel files now provided for Table 1 and the Expanded View Tables.

“Data described in submitted manuscripts should be deposited in a MIAME-compliant format with one of the public databases. We would therefore ask you to submit your microarray data to the ArrayExpress database maintained by the European Bioinformatics Institute for example. ArrayExpress allows authors to submit their data to a confidential section of the database, where they can be put on hold until the time of publication of the corresponding manuscript. Please see <http://www.ebi.ac.uk/arrayexpress/Submissions/> or contact the support team at arrayexpress@ebi.ac.uk for further information.”

DONE. Copy number array submitted to ArrayExpress (accession no. E-MTAB-5450).

“For experiments involving human subjects the authors must identify the committee approving the experiments and include a statement that informed consent was obtained from all subjects and that the experiments conformed to the principles set out in the WMA Declaration of Helsinki [<http://www.wma.net/en/30publications/10policies/b3/>] and the NIH Belmont Report [<http://ohsr.od.nih.gov/guidelines/belmont.html>]. Any restrictions on the availability or on the use of human data or samples should be clearly specified in the manuscript. Any restrictions that may detract from the overall impact of a study or undermine its reproducibility will be taken into account in the editorial decision. In this case I refer specifically to the data obtained from the blood and bone marrow samples taken at diagnosis from patients with pediatric ALL. We note that this was not mentioned in the checklist either. If applicable the checklist should also be amended and the new version uploaded with the revised manuscript.”

DONE. Statement added in Materials and Methods (page 18-19). Checklist updated as requested.

“You are welcome to suggest a striking image or visual abstract to illustrate your article. If you do, please provide a jpeg file 550 px-wide x 400-px high”

DONE. Image abstract provided in the requested format.

Corresponding Author Name: Björn Nilsson (bjorn.nilsson@med.lu.se)

Manuscript Number: EMM-2016-06660